# Xylanase Production by *Talaromyces amestolkiae* Valuing Agroindustrial Byproducts

**DOI:** 10.3390/biotech11020015

**Published:** 2022-05-17

**Authors:** Giórgia S. Barbieri, Heitor B. S. Bento, Fernanda de Oliveira, Flávio P. Picheli, Lídia M. Dias, Fernando Masarin, Valéria C. Santos-Ebinuma

**Affiliations:** Department of Engineering Bioprocess and Biotechnology, School of Pharmaceutical Sciences, São Paulo State University (UNESP), Rodovia Araraquara-Jau km. 01, Araraquara 14800-903, SP, Brazil; giorgia.barbieri@unesp.br (G.S.B.); heitor.bento@unesp.br (H.B.S.B.); fer.oliveiractb@gmail.com (F.d.O.); flavio.picheli@unesp.br (F.P.P.); lidia_manfrin@yahoo.com.br (L.M.D.); fernando.masarin@unesp.br (F.M.)

**Keywords:** xylanase, *Talaromyces amestolkiae*, enzymatic hydrolysis, submerged cultivation, agroindustrial byproducts

## Abstract

In general, agroindustrial byproducts can be easily assimilated by several microorganisms due to their composition, which is rich in carbohydrates. Therefore, they could be appropriate for use as raw materials in a sustainable refinery concept, including the production of hydrolytic enzymes with industrial applicability. In this work, xylanase production by the filamentous fungi *Talaromyces amestolkiae* in submerged culture was evaluated using five agroindustrial byproducts, namely, wheat bran, citrus pulp, rice bran, peanut skin, and peanut shell. Firstly, the aforementioned byproducts were characterized in terms of cellulose, xylan, lignin, and extractives. Next, production studies were performed, and wheat bran generated the highest enzymatic activity (5.4 U·mL^−1^), probably because of its large amount of xylan. Subsequently, a factorial design was performed to evaluate the independent variables yeast extract, wheat bran, K_2_HPO_4_, and pH, aiming to improve the variable response, xylanase activity. The condition that promoted the highest production, 13.02 U·mL^−1^ (141% higher than the initial condition), was 20 g·L^−1^ wheat bran, 2.5 g·L^−1^ yeast extract, 3 g·L^−1^ K_2_HPO_4_, and pH 7. Thus, industrial byproducts with a high content of xylan can be used as a culture medium to produce xylanase enzymes with a *Talaromyces* strain through an economical and sustainable approach.

## 1. Introduction

The waste fraction of byproducts generated by agroindustrial activities are mainly composed of biomass, generally obtained from vegetal sources [1]. In Brazil, approximately 200 million tons of agroindustrial waste is produced per year. The sources are greatly diversified, including residues from agriculture and industrial processes such as sugarcane straw and bagasse, forest residues, rice husk, bark, and wheat bran, among others [2]. When these residues are disposed into the environment, they can cause several impacts. For example, they can be sent to landfills and be degraded by microorganisms or incinerated, generating the emission of greenhouse gases into the atmosphere; they can pollute soil and water and cause social impacts [3]. Since these residues are mostly vegetal biomass, their composition is predominantly lignocellulose, a complex polymeric material that consists mainly of cellulose (30–60%) followed by hemicellulose (20–40%) and lignin (10–20%) [4,5]

The most common way to turn the lignocellulosic byproducts into added value substances is by acid hydrolysis, although they need to react with mineral acids, e.g., sulfuric, phosphoric, hydrochloric, acetic, and nitric acids being necessary for their treatment, before being disposed in order to avoid environmental impact [6]. The same is true for the alkaline transformation of biomass, which can turn these wastes into added-value products, however, it also employs chemical substances that need treatment, such as ammonium, potassium, calcium, and sodium hydroxides [7]. Waste biorefineries are diversified power plants that allow the conversion of residues and byproducts into value-added products, such as fertilizers, biofuels, and value-added chemicals, making them a convenient strategy to reduce the disposal of waste into watercourses or into the soil [8].

An interesting industrial class of valuable products is enzymes—biocatalysts with diverse functional properties able to accelerate biochemical reactions through the bioconversion of substrates into products, as well as modulate metabolic pathways. Enzymes are also important to make some industrial processes viable, including the degradation of lignocellulose biomass by specific enzymes such as xylanases [9,10]. This class of enzymes can be produced by vegetables, microorganisms (bacteria, algae, fungi, and yeast), protozoans, and even crustaceans. Xylanases are able to cleave xylan, the second-most abundant known polysaccharide (second to cellulose) [11] and are useful for multiple purposes, such as the production of paper, textiles, pharmaceuticals, and foods [12]. Even so, filamentous fungi are the most suited microorganism producers of xylanase due to their capacity to generate large amounts of extracellular enzymes using a wide range of low-cost substrates, such as waste and byproducts. Among the filamentous fungi, *Aspergillus*, *Trichoderma*, and *Talaromyces* genera, which include the *Talaromyces amestolkiae* (previously called *Penicillium purpurogenum*) are described in the literature as enzyme producers [13,14,15].

The genus *Talaromyces* has an ascomata with a soft cell wall that is covered with interwoven hyphae. In the past, they were considered part of the *Penicillium* subgenus *Biverticillium*. However, they are now considered another genus due to some morphological, physiological, and molecular differences. These microorganisms are characterized by forming dark gray or gray-green colonies with mycelium with colors from pink to yellow. They are also able to produce red pigments [16,17]. *Talaromyces* are also capable of synthesizing xylanase (EC, 3.2.1.8) and β-xylosidase (EC, 3.2.1.37), both xylanolytic enzymes that generate xylooligosaccharides (XOS) from hemicellulose [18,19].

There are two main ways to produce enzymes from microorganisms: solid- and submerged-state cultivations. Solid-state is easier to scale up and has no apparent free water in the process, though it possesses sufficient moisture for the fungus to grow using lignocellulosic material as the substrate, avoiding some downstream steps such as dewatering [20]. Submerged cultivation, in turn, has a higher yield, an accelerated production rate, is simpler to control, and is more effective for fungi that require a high moisture content in order to grow [21].

Thus, the objective of this study was to evaluate five agroindustrial residues, namely, wheat bran, rice bran, peanut husk, peanut skin, and citrus pulp, for the production of xylanases by submerged cultivation of *T. amestolkiae* in order to propose a feasible waste valorization route for low-cost xylanolytic enzyme production.

## 2. Materials and Methods

### 2.1. Materials

The agroindustrial byproducts wheat bran and rice bran were obtained from the local market, peanut skin and peanut shell were donated by Jazam Peanuts (Pompéia, SP, Brazil), and citrus pulp was donated by Cutrale (Araraquara, SP, Brazil). The other components used were of analytical grade.

### 2.2. Agroindustrial Byproducts Characterization

The analysis of the composition of agroindustrial residue in terms of carbon and nitrogen content was performed by the Chemistry Institute from University of São Paulo (IQ-USP, São Paulo, Brazil). Thereafter, cellulose, xylan, total lignin, and extracts were studied according to the following methodology. The chemical composition of all byproducts was determined by subsequent extractions. First, 3 g of material was placed in filter paper envelopes and was subjected to solid–liquid extraction using hexane as extractor for 6 h in the Soxhlet apparatus (Laborglas, São Paulo, Brazil). The solvent was evaporated using a Heidolph-Hei VAP-Advantage rotary evaporator (Schwabach, Germany) at 40 °C, 150 rpm, and 250 bar, the sample was dried in an oven at 50 °C for 12 h and the mass (lipid content) was measured by gravimetry. Similarly, the remaining extractive content was subjected to ethanol 95% (*v*/*v*) for 6 h in the same Soxhlet apparatus (Laborglas, São Paulo, Brazil) for extractive determination. The remaining material was washed with distilled water, dried, and weighed. Then, 0.3 g of the byproduct was mixed with 3 mL of H_2_SO_4_ 72% (*w*/*w*) for 1 h at 30 °C [22,23]. The hydrolysate was filtered using porous glass filters and the retained material was washed with 10 mL of water and dried over 12 h. The filtrate was used to determine structural carbohydrates by high performance liquid chromatography (HPLC) with a refractive index detector (Shimadzu). The samples were passed through a 0.45 µm membrane and were injected into the HPLC system coupled with a RID detector (Shimadzu-LC 2 D). The analyses were performed using an HPX-87H column (Biorad Aminex) with H_2_SO_4_ 5 mM with a flow rate 0.6 mL min^−1^ and a column oven at 60 °C. Sample injection volume was 20 µL. Cellulose and hemicellulose contents in the samples were calculated from glucose, arabinose, xylose, and acetic acid data.

### 2.3. Microorganism

*T. amestolkiae* (DPUA 1275) was generously provided by the Culture Collection of the Federal University of Amazonas (DPUA), AM. The inoculum was prepared in PDA plates supplemented with yeast extract (yeast extract 5 g·L^−1^, potato dextrose agar 39 g·L^−1^ and the cultures were maintained at 30 °C for 168 h.

### 2.4. Xylanase Production

For the submerged cultivation, the different agricultural byproducts were used as main carbon source substrate. Five mycelial agar discs (8 mm diameter) of *T. amestolkiae* were transferred to 50 mL of submerged culture medium in 250 mL Erlenmeyer flasks. Initially, the culture medium was composed of (g·L^−1^): agroindustrial byproduct (10), yeast extract (5), and K_2_HPO_4_ (1) at pH 7. The cultivation process was performed in an orbital shaker incubator at 30 °C and 150 rpm for 168 h. At the end of the cultivation, the culture media were recuperated by filtration using Whatman No. 1 filter paper and then centrifuged at 2000× *g* for 15 min. The obtained supernatants were used for enzymatic activity analysis.

### 2.5. Cultivation Parameters Optimization

A 2^4^ factorial design was performed with 4 central points varying the independent variables—concentrations of wheat bran, yeast extract, and K_2_HPO_4_ and pH - xylanase activity was considered as the variable responses. The other variables were maintained at a constant level. The confidence level considered was 95%. Software Statistica 13.0 (Statsoft) was used to determine the effect of the analyzed factors. The optimum conditions obtained by the factorial design were applied to analyze the cultivation period.

### 2.6. Enzymatic Activity Assay

Xylanase activity was determined by the xylan hydrolysis reaction. The reaction mixture was composed of 250 µL of xylan Larchwood 1% (*w*/*v*) in 50 mM of citrate buffer at pH 5.0 plus 250 µL of enzymatic solution (fermentation supernatant after filtration and centrifugation) carried for 20 min at 50 °C [24]. The amount of xylose was determined by measuring the release of reducing sugar from the reactional medium using 3,5- dinitrosalicylic acid (DNS) [25]. The measurements were performed using an EnSpire Alpha Plate Reader spectrophotometer (PerkinElmer^®^, Waltham, MA, USA) at 540 nm. One unit of xylanase activity was defined as the amount of reducing sugar (xylose) released per minute per milliliter of enzyme extract in the experimental conditions.

### 2.7. Enzyme Characterization

#### 2.7.1. Determination of Optimum pH and Temperature

In order to evaluate the optimal pH of the enzyme, Mcllvaine buffer [26] was used at different pH values: 3.0, 4.0, 5.0, 6.0, 7.0, and 8.0, and the lyophilized extract was used at 0.01% *w*/*v*. This assay was carried out at 50 °C. To determine optimum temperature, the reaction occurred using the lyophilized extract (0.01% *w*/*v*) in Mcllvaine buffer, incubating the solution at 30, 40, 50, 60, 70, and 80 °C in a New Ethics model 521/2DE thermostatic bath (Piracicaba, SP, Brazil). Both assays were performed for 20 min at 50 °C according to the methodology described previously (Section 2.6).

#### 2.7.2. Enzymatic Stability in Function of pH and Temperature

To study the influence of pH on the activity of the xylanase produced, assays were performed using Mcllvaine buffer at different pH value: 3.0, 4.0, 5.0, 6.0, 7.0, and 8.0 at 30 °C for 24 h. The influence of temperature on xylanase activity was studied in the range from 30 to 60 °C in a thermostatic bath New Ethics model 521/2DE (Piracicaba, São Paulo, Brazil) using the optimal pH obtained in the previous analysis for 24 h. The exposed enzyme aliquots were used to determine the residual activity at defined times.

### 2.8. Data Analysis and Presentation

All experiments were performed in triplicate. The results are presented by the means and standard deviations.

## 3. Results

### 3.1. Agroindustrial Byproducts Characterization

The characterization of complex substrates as agroindustrial byproducts, is essential to contributing to the deep understanding of fermentative processes, enabling deeper scientific insights into the bioprocess’s challenges and potentials.

The composition of the five byproducts is shown in Table 1. It is shown that wheat bran and citrus pulp presented the most elevated cellulose and xylan contents (32.7% and 33.3% of cellulose and 14.6% and 10.3% of xylan, for wheat bran and citrus pulp, respectively) when compared to the other byproducts. The higher lignin content in the rice bran and peanut shell samples (24.6% and 43.6%, respectively) may cause an inhibitory effect and subsequent efficiency losses in the fermentative processes. In this same context, analogously, the byproducts with the lowest lignin content (wheat bran and citrus pulp) present potential for application in the cultivation process as nutrient sources for microorganisms.

In all experiments, the total of compounds was higher than 90%, except the citrus pulp. Other undetermined components necessary to reach 100% could be ash or methyl glucuronic acid, which were not analyzed in this work. However, these results highlight the elevated carbon content in all byproducts analyzed that could be employed to generate bioproducts with high value aggregated, such as enzymes.

### 3.2. Xylanase Production

The xylanase production screening was performed in the cultivation of *T. amestolkiae* in a culture medium containing the agroindustrial byproducts wheat bran, rice bran, citrus pulp, peanut skin, and peanut shell as carbon sources. The results obtained are expressed in Figure 1 in the function of enzymatic activity (U·mL^−1^). The results indicated significantly higher xylanase activity for the medium composed of wheat bran when compared to the other byproducts (almost 25-fold higher than the second-highest enzymatic activity obtained in all tested media formulations).

These results may be intrinsically related to the byproducts and subsequent media composition, whereas the medium containing wheat bran promoted an adequate and balanced composition for xylanase production by *T. amestolkiae*. This adequate composition is possibly related to the elevated xylan and low lignin contents and balanced Carbon/Nitrogen ratio. Besides these important parameters, the presence of non-quantified specific minor nutrients, such as minerals, may also have influenced the results obtained.

Here, xylanase activity achieved up to 5.41 U·mL^−1^ when wheat bran was applied as a carbon source.

The results indicated the influence of the factors analyzed since a wide range of enzymatic activity was obtained (from 0.32 to 13.02 U·mL^−1^). From the results, it is possible to infer that alkaline medium (pH 9.0) was not favorable for xylanase production presenting enzymatic activities in the range of 0.32–3.38 U·mL^−1^. The highest xylanase activity was observed in the experimental design central point (20 g·L^−1^ wheat bran, 2.5 g·L^−1^ yeast extract, 3 g·L^−1^ K_2_HPO_4_, and pH 7.0), achieving approximately 13.02 U·mL^−1^.

In order to elucidate the statistical influence of the studied variable factors, the Pareto chart of effects is shown in Figure 2. The diagram indicates that the variable pH and the combination of pH and yeast extract were significant under 95% confidence level. The negative effect of pH indicates once again that an increase in the alkalinity of the medium causes a decrease in enzyme production. On the other hand, the negative interaction effect of yeast extract x pH indicates that higher concentrations of yeast extract in lower pH medium have the potential to increase xylanase production by *T. amestolkiae* cultivated in wheat-bran-based culture media. Additionally, yeast extract presents a complex amino acid, vitamin, and mineral composition that may be essential for the production of proteins [27].

The central points presented the highest enzymatic activities, around 13.02 U·mL^−1^, which correspond to 141% higher than the activity previously obtained using the standard medium. So, this cultivation condition was selected for the following experiments.

Another important parameter to be taken into consideration in the optimization of production processes is the microorganism cultivation period, i.e., the cultivation time. This parameter was evaluated according the xylanase production as a function of enzymatic activity over cultivation time up to 240 h is showed in Figure 3.

### 3.3. Ideal Conditions for Xylanase Production

Since wheat-bran-based cultivation medium showed far better results than the other analyzed byproducts, it was selected for deeper studies. In order to evaluate the influence of some cultivation parameters on xylanase production, a full 2^4^ factorial design with four central points was applied to analyze the effects of medium component concentration (wheat bran, yeast extract, and K_2_HPO_4_) and pH. The variables levels in real values as the results as functions of enzymatic activity are indicated in Table 2.

The results depicted in Figure 3 indicate the highest xylanase production in 120–144 h of cultivation. The profile of the enzymatic activity over cultivation time is probably related to a period of induction of enzyme production by the substrate’s presence (xylan) for about 72 h, where a significative increase in the activity can be noticed achieving its maximum at 144 h. The subsequent decrease in enzymatic activity may be caused by inhibition of some product formed during biomass growth. Additionally, the lower xylan induction is due to its consumption during the cultivation period and a higher exposition of the lignin and possible inhibitory phenolic derivatives. The results after 120 and 144 h did not show significative difference considering the Tukey test. Considering time is a key factor for the industrial establishment of a bioprocess and is intrinsically related to process costs, 120 h of cultivation can be considered the best time for industrial production purposes.

### 3.4. Xylanase Characterization

Enzyme characteristics such as optimum pH and temperature with several pH values and temperatures for stability are essential information for enzyme application. In this sense, the xylanase produced was characterized in relation to optimum pH and temperature (Figure 4) with different pH and temperatures for stability (Figure 5). Figure 4A shows xylanase activities over a pH range from 3.0 to 8.0 and indicates an optimum pH of 4.0. The optimum temperature was studied in the range from 30 to 80 °C, and the results indicate a higher catalytic activity in the range from 50 to 60 °C (Figure 4B).

Enzyme stability is an important parameter set to indicate the feasibility of the target enzyme production and application. Xylanase stability was evaluated by incubation of the cultivation medium containing the produced enzyme at different pH and temperatures for up to 24 h. Figure 5A,B shows the relative activity over incubation time at different pH and temperature, respectively. The samples presented similar residual activity in the range of pH 4.0–8.0 (15.4–31.5%) after 24 h of incubation. A lower activity value was observed at pH 3.0, indicating that an extremely acidic medium may cause enzyme denaturation and consequently activity loss. The results indicated that the produced enzyme is able to act in a wide range of pH, making it able to be applied in the product of acidic and alkaline lignocellulosic material pretreatments after light neutralization.

Regarding temperature stability shown in Figure 5B, it is possible to notice that the higher the temperature, the lower the relative enzymatic activity after 24 h of incubation. However, it is interesting to point out that in the first 6 h of incubation, the enzyme demonstrated that it was more active at 40 °C, presenting 49% of the initial activity after 6 h of incubation, while 29.2 and 14.7% of the initial activity were observed for 50 and 60 °C, respectively.

## 4. Discussion

From the byproduct results of chemical composition (Table 1), it was evident that wheat bran contains a higher proportion of xylan when compared to the other residues studied. This indicates wheat bran as a potential source for cellulolytic and xylanolytic enzyme production by microorganisms since the presence of these xylan compounds may induce the corresponding enzyme production [29]. Additionally, the inhibitory lignin derivatives and some substrate accessibility difficulties may also occur in the process due to structural hindrances [30,31]. The use of wheat bran and similar xylan-based materials for xylanase production by filamentous fungi is widely reported by other authors, as depicted in Table 3.

As can be seen in Table 3, several authors have reported similar results in xylanase production, presenting enzymatic activity from 7.5 to 24 U·mL^−1^, optimum pH in the range from 4.5 to 8.0, and optimum temperature between 50 and 80 °C. These results demonstrate that xylanase production from wheat-bran-based culture media presents a very versatile range of application conditions, being able to be applied in different industrial bioprocesses. Walia et al. (2017) [44] have reported that bacterial xylanase usually presents optimum activity at neutral pH, while fungal xylanase generally is more active in slightly acidic pH. The authors also state that the optimum temperature is usually found at mesophilic temperatures from 40 to 60 °C, which is in accordance with the results shown in Figure 5 that indicate higher catalytic activity in the range from 50 to 60 °C.

The best biotechnological parameters found in the present research from the statistical design optimization shown in Table 2 and Figure 2; Figure 4 are also in accordance with the literature, which has indicated that slightly acidic culture media are more adequate for filamentous fungi cultivation [45]. Although the enzyme was not purified, SDS-PAGE electrophoresis showed a band between 25 and 35 kDa (Appendix A), which is in accordance with the molecular weight presented to other xylanases reported in the literature, as presented in Table 3.

It is well highlighted in the literature that the use and valorization of low-cost and abundant agroindustrial byproducts is a key factor for the feasibility of establishing fungi-based biorefineries, contributing not only to operational cost reduction, but also to solid waste management and subsequent environmental protection. The results showed and discussed in this present study may provide an essential basis in the establishment of a complex integrated network of processes in a biorefinery context by inserting a high value-added enzyme production chain from low-cost byproducts.

## 5. Conclusions

The production of xylanase enzymes by *T. amestolkiae* using low-cost agroindustrial lignocellulosic byproduct, namely wheat bran, was studied. Possibly, the high amount of xylan in this byproduct is the main factor responsible for enzyme production, which achieved 14.25 U·mL^−1^ of xylanase activity in the best conditions for cultivation. The produced enzyme showed good stability in a wide range of pH and temperature. The results demonstrated herein indicate that filamentous fungus *T. amestolkiae* can be used as an efficient tool in the valorization of lignocellulosic byproducts by the production of high-value hemicellulolytic enzymes, such as xylanases.

## Figures and Tables

**Figure 1 biotech-11-00015-f001:**
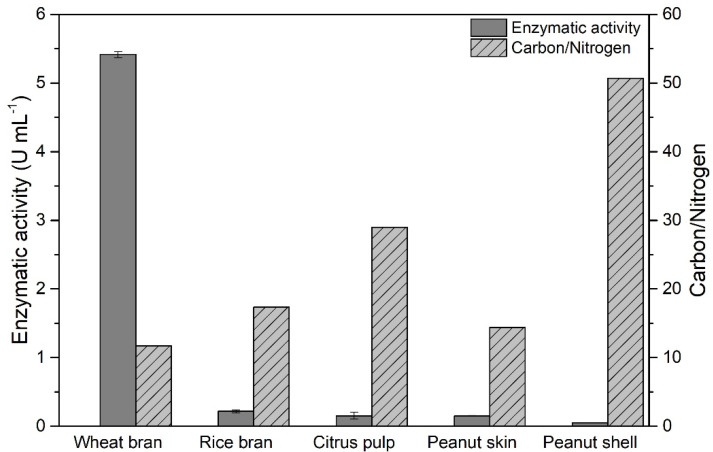
Production of xylanase enzyme by the cultivation of *T. amestolkiae* in an orbital shaker incubator at 30 °C and 150 rpm for 168 h using different agroindustrial byproducts. The error bars represent the standard deviation of triplicates.

**Figure 2 biotech-11-00015-f002:**
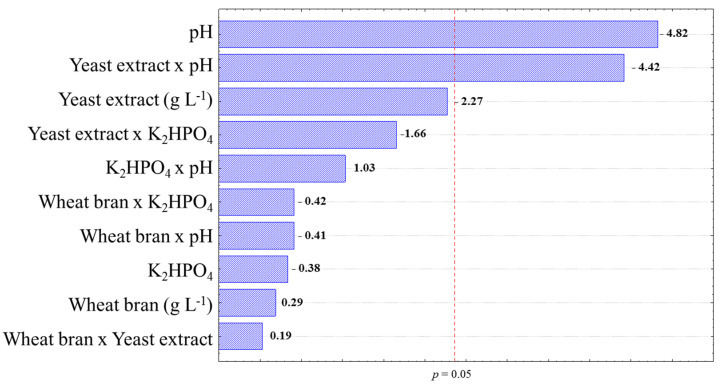
Pareto chart of the 2^4^ factorial design for studying xylanase production by *T. amestolkiae* in an orbital shaker incubator at 30 °C and 150 rpm for 168 h using wheat bran. The line in the chart represents a reference line; any factor that extends past this line is of significant effect at *p*-value < 0.05 [28].

**Figure 3 biotech-11-00015-f003:**
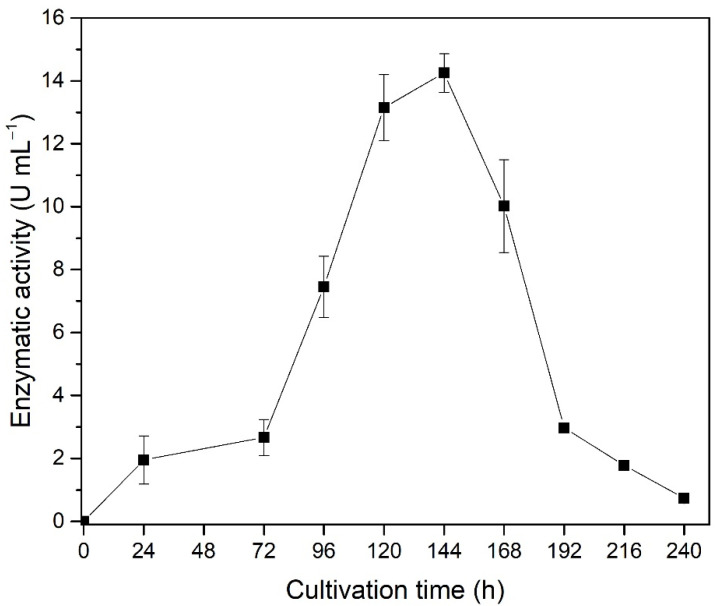
Production of xylanase by the cultivation of *T. amestolkiae* in an orbital shaker incubator at 30 °C and 150 rpm over 240 h using wheat bran as agroindustrial byproducts. The error bars represent the standard deviation of triplicates.

**Figure 4 biotech-11-00015-f004:**
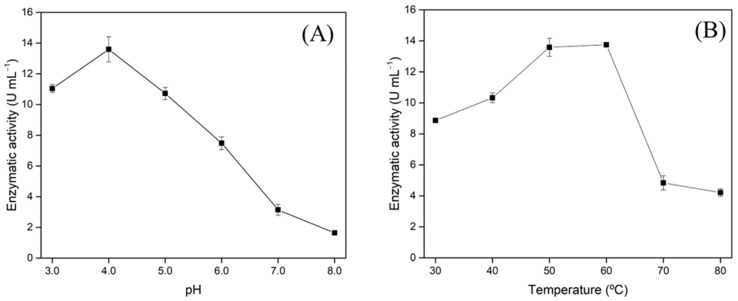
Characterization of lyophilized xylanase produced by cultivation of *T. amestolkiae* in an orbital shaker incubator at 30 °C and 150 rpm for 240 h using wheat bran as agroindustrial byproducts in relation to optimal pH (**A**) and temperature (**B**). The error bars represent the standard deviation of triplicates.

**Figure 5 biotech-11-00015-f005:**
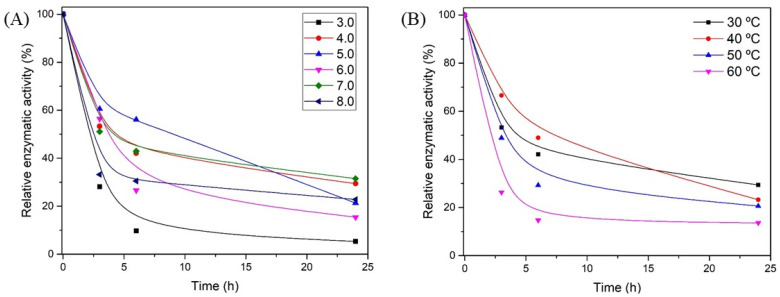
Relative activity of xylanase produced by submerged cultivation of *T. amestolkiae* using wheat bran as substrate as a function of time at different pH (**A**) and temperature (**B**). The incubation temperature was maintained at 30 °C in the studies of pH stability. In the studies of temperature stability, the pH was maintained at 4.0. Nonvisible standard deviations indicate that the marker is greater than the deviation itself. Results expressed as a percentage of activity in relation to the control (zero time).

**Table 1 biotech-11-00015-t001:** Component characterization of agroindustrial byproducts applied as a substrate for filamentous fungus cultivation.

Component (% wt.)	Wheat Bran	Rice Bran	Citrus Pulp	Peanut Skin	Peanut Shell
Cellulose	32.7	25.4	33.3	5.0	20.9
Xylan	14.6	6.5	10.3	1.1	9.8
Lignin	8.1	24.6	7.5	14.5	43.6
Extractives	14.0	26.3	24.5	42.1	3.6
Oil	22.6	10.2	9.9	35.3	20.1
Total	92	93	85.5	98	98

**Table 2 biotech-11-00015-t002:** Factorial design 2^4^ with four central points aiming at xylanase production using *T. amestolkiae*.

Run	Wheat Bran (g·L^−1^)	Yeast Extract (g·L^−1^)	K_2_HPO_4_ (g·L^−1^)	pH	Enzyme Activity (U·mL^−1^)
1	10	0	1	5	1.34 ± 0.16
2	30	0	1	5	2.57 ± 0.37
3	10	5	1	5	8.67 ± 0.26
4	30	5	1	5	8.33 ± 0.70
5	10	0	5	5	2.48 ± 0.25
6	30	0	5	5	2.80 ± 0.21
7	10	5	5	5	5.55 ± 1.02
8	30	5	5	5	6.09 ± 0.51
9	10	0	1	9	1.16 ± 0.07
10	30	0	1	9	1.61 ± 0.14
11	10	5	1	9	0.35 ± 0.02
12	30	5	1	9	0.76 ± 0.11
13	10	0	5	9	3.38 ± 0.32
14	30	0	5	9	1.85 ± 0.03
15	10	5	5	9	0.32 ± 0.03
16	30	5	5	9	0.35 ± 0.02
17–20 *	20	2.5	3	7	13.02 ± 0.88

* Central points.

**Table 3 biotech-11-00015-t003:** Comparison of xylanase production by fungi from wheat-bran- and xylan-based materials in the literature.

Microorganism	Main Carbon Source	Enzymatic Activity (U·mL^−1^)	Molecular Weight (kDa)	Optimum pH	Optimum T (°C)	References
*Aspergillus niger*	Wheat bran	24	30	7.5	60	[32]
*Aspergillus fischeri* Fxn1	Wheat bran	ND	ND	6.0	60	[33]
*Aspergillus fumigatus* AR1	Wheat bran	17.5	212–253	6–6.5	60	[34]
*Aspergillus fumigatus* RSP-8 (MTCC 12039)	Wheat bran	≈22	ND	ND	ND	[35]
*Cephalosporium* sp.	Wheat bran	ND	35	7.5	50	[36]
*Penicillium oxalicum* ZH-30	Wheat bran	14.5	ND	ND	50–60	[37]
*Penicillium sclerotiorum*	Oat spelts xylan	7.8	ND	4.5	50	[38]
*Talaromyces byssochlamydoides* YH-50	Wheat bran	ND	ND	5.5	70	[39]
*Talaromyces emersonii* (IMI 392299)	Wheat bran	ND	ND	4.5–5.0	70	[40]
*Talaromyces stollii* LV186	Corn stover	7.5	ND	ND	ND	[41]
*Talaromyces thermophilus*	Oat spelts xylan; Wheat bran; rabbit food	ND	25	7.0–8.0	75–80	[42]
*Tuber maculatum*	Beechwood xylan	13.15	ND	5.0	50	[43]

## Data Availability

Not applicable.

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
