# Peer review of "Xylanase Production by Talaromyces amestolkiae Valuing Agroindustrial Byproducts"

_biotech, 2022, doi:10.3390/biotech11020015_

Round 1

Reviewer 1 Report

The result of electrophoresis does not clearly demonstrate the presence of enzymes. An SDS-PAGE demonstrates the presence of protein. It is necessary to present some other technique, such as zymograms or eliminate the section.

Author Response

Reviewer #1:

The result of electrophoresis does not clearly demonstrate the presence of enzymes. An SDS-PAGE demonstrates the presence of protein. It is necessary to present some other technique, such as zymograms or eliminate the section.

RESPONSE: We thank the reviewer for the observation, the authors agreed and have removed this section from the manuscript.

Reviewer 2 Report

In publication application of agroindustrial byproducts in production of xylanase by Talaromyces amestolkiae was described.This publication seems to be within the scope of journal. However it needs several corrections to be more acceptable for publication.

  1. Lines 66 and 67: Please add EC number of both enzymes.
  2. The introduction should take into account last review about Talaromyces amestolkiae glycosyl hydrolases: Prieto, A., de Eugenio, L., Méndez-Líter, J. A., Nieto-Domínguez, M., Murgiondo, C., Barriuso, J., et al. Fungal glycosyl hydrolases for sustainable plant biomass valorization: Talaromyces amestolkiae as a model fungus. International Microbiology, 2021, 1-14.
  3. Line 102: Please add information about manufacturer and type of HPLC system.
  4. Line 103: Please correct “with flow rate 0.6 mL/min”. What was the injection volume? Please add appropriate information.
  5. Line 107: Please add reference number of Talaromyces amestolkiae in collection of microorganisms.
  6. Line 131: Please add information: What was the enzyme dilution? What buffer was used to dilute the enzyme?
  7. Line 133: Please correct name of compound. What is the position of the nitro groups on the benzene ring?Is it about 3,5- dinitrosalicylic acid (DNS)? Please add information in publication: Which plate reader was used and at what wavelength was taken measeruments.
  8. Please correct units in whole manuscript: e.g. “g·L-1” instead of “L-1” or “U·mL-1” instead of “U.mL-1”.
  9. Please add information about amount of experiments repetitions.
  10. References must be formatted according to instruction for authors.

Author Response

Reviewer #2:

In publication application of agroindustrial byproducts in production of xylanase by Talaromyces amestolkiae was described. This publication seems to be within the scope of journal. However it needs several corrections to be more acceptable for publication.

  • Lines 66 and 67: Please add EC number of both enzymes.

RESPONSE: Thank you for the suggestion, EC numbers were added in lines 66 and 67 as follows:

“xylanase (EC, 3.2.1.8) and β-xylosidase (EC, 3.2.1.37)”

  • The introduction should take into account last review about Talaromyces amestolkiae glycosyl hydrolases: Prieto, A., de Eugenio, L., Méndez-Líter, J. A., Nieto-Domínguez, M., Murgiondo, C., Barriuso, J., et al. Fungal glycosyl hydrolases for sustainable plant biomass valorization: Talaromyces amestolkiae as a model fungus. International Microbiology, 2021, 1-14.

RESPONSE: Thank you for the suggestion, the reference was added in the revised manuscript as reference 18 (Line 69)

  • Line 102: Please add information about manufacturer and type of HPLC system.

RESPONSE: The information has been added in line 102 as follows: “HPLC system coupled with a RID detector (Shimadzu - LC 20AD)”

  • Line 103: Please correct “with flow rate 0.6 mL/min”. What was the injection volume? Please add appropriate information.

RESPONSE: Thank you for the correction, the information was rewritten in lines 103-104: “with flow rate 0.6 mL min-1 … Sample injection volume was 20 µL.”

  • Line 107: Please add reference number of Talaromyces amestolkiae in collection of microorganisms.

RESPONSE: We apologize by our mistake. The reference number of Talaromyces amestolkiae, namely DPUA 1275, was included in the line 108.

  • Line 131: Please add information: What was the enzyme dilution? What buffer was used to dilute the enzyme?

RESPONSE: The phrase was rewritten in order to make the information clearer. The enzymatic dilution actually meant the enzymatic solution composed by the fermentation supernatant.

  • Line 133: Please correct name of compound. What is the position of the nitro groups on the benzene ring? Is it about 3,5- dinitrosalicylic acid (DNS)? Please add information in publication: Which plate reader was used and at what wavelength was taken measurements.

RESPONSE: Thank you for the correction. The information was corrected as follows: “The amount of xylose was determined by measuring the release of reducing sugar from the reactional medium using 3,5- dinitrosalicylic acid (DNS) [24]. The measurements were performed using an EnSpire Alpha Plate Reader spectrophotometer (PerkinElmer®) at 540 nm.”

  • Please correct units in whole manuscript: e.g. “g·L-1” instead of “L-1” or “UmL-1” instead of “U.mL-1”.

RESPONSE: Thank you for the correction, the whole manuscript was revised and the corrections were made as suggested.

  • Please add information about amount of experiments repetitions.

RESPONSE: All experiments were performed in triplicate, this information was added in lines 155-156.

  • References must be formatted according to instruction for authors.

RESPONSE: The authors acknowledge once again the efforts done in the revision; the references were formatted according the instructions in the revised manuscript.

Reviewer 3 Report

The manuscript from Barbieri et al., discusses the effect of different parameters on the enzymatic activity of xylanase. Further the authors have stated xylanase activity on five agro industrial byproducts. Although the study results look convincing but at many places the conclusion appears overshooting and lacks appropriate controls. Here are my comments to improve the manuscript:

  • My biggest complaint with manuscript is that it is poorly written and because of which the reader cannot understand the actual message the authors want to convey, and it is difficult to understand How the experiments were performed? Throughout the manuscript sentences are not clear. For example, Line 12 – 13, Line 45 – 46, Line 52 – 53, Line 192 – 195.
  • The language is nonscientific and not in accordance with publishing standards.
  • It is not clear how many times the experiments were performed. If experiments were conducted in replicates or just a single time analysis was performed? No statistical analysis information is provided. Statistical significance of these results is not clear.
  • Figures could be more informative and organized. Legends are also not informative. For example, Figure 6 the protein bands are not clearly visible.
  • The Material and method section is not appropriately described to replicate the experiments. For example, Line 104 to 105, the author mentioned they calculated hemicellulose content but did not mention How? Line 92, what % of hexane?, Line 134 what mL of enzyme? Line 138 to 153, (section 2.7.1 and 2.7.2 are these same)? SDS PAGE procedure is incomplete
  • The authors have mainly performed enzymatic activity probably on crude extract which is not clear How it was prepared. They have tested different pH, different temperatures and stability of xylanase but it is not clear How these experiments were performed and why the authors have not purified the xylanase to test these parameters which is a usual method for enzyme characterization?
  • Line 192 to 195: The authors mentioned “The presence of non-quantified specific minor nutrients ………..” How the authors reached to this conclusion? Did they perform any experiment for the same?
  • Line 202 to 207: Which results the authors are trying to show for mentioned text? Does central condition is the term used in literature or authors used it for the first time as it seems inappropriate.
  • Line 226-229: The authors ended the statements abruptly and continued later as part of different results.
  • Xylanase activity is higher at high temperature. Does it mean it can be stored at high temperature? Because it appears xylanase should have high stability in that case.
  • Line 296- 298: Not clear what authors want to convey and SDS is not informative.
  • I will recommend the authors to take help from native English speaker for data interpretation and presentation.

Author Response

Reviewer #3: The manuscript from Barbieri et al., discusses the effect of different parameters on the enzymatic activity of xylanase. Further the authors have stated xylanase activity on five agro industrial byproducts. Although the study results look convincing but at many places the conclusion appears overshooting and lacks appropriate controls. Here are my comments to improve the manuscript:

  • My biggest complaint with manuscript is that it is poorly written and because of which the reader cannot understand the actual message the authors want to convey, and it is difficult to understand How the experiments were performed? Throughout the manuscript sentences are not clear. For example, Line 12 – 13, Line 45 – 46, Line 52 – 53, Line 192 – 195.

RESPONSE: Thank you for your revision. The whole manuscript was revised, specifically the aforementioned sentences were rewritten in order to make it clearer for the readers. They are highlighted in yellow in the revised manuscript.

  • The language is nonscientific and not in accordance with publishing standards.

RESPONSE: The manuscript was carefully revised. The authors believe that the revised manuscript presents an improved scientific language by the incorporation of the reviewers’ corrections and suggestions. Thank you again for your revision.

  • It is not clear how many times the experiments were performed. If experiments were conducted in replicates or just a single time analysis was performed? No statistical analysis information is provided. Statistical significance of these results is not clear.

RESPONSE: Thank you for your observation. The experiments were performed in triplicates and a section (2.8) was inserted in the manuscript to make the data analysis clearer for the readers.

  • Figures could be more informative and organized. Legends are also not informative. For example, Figure 6 the protein bands are not clearly visible.

RESPONSE: The figures legends were revised. Moreover, as also suggested by reviewer #1 figure 6 was removed.

  • The Material and method section is not appropriately described to replicate the experiments. For example, Line 104 to 105, the author mentioned they calculated hemicellulose content but did not mention How? Line 92, what % of hexane? Line 134 what mL of enzyme? Line 138 to 153, (section 2.7.1 and 2.7.2 are these same)? SDS PAGE procedure is incomplete.

RESPONSE: The authors acknowledge the revisor comments. Several changes were performed in the revised manuscript. Section 2.7.1 and 2.7.2 are not the same, the first one is regarding the determination of the optimum enzymatic acting conditions and section 2.7.2 is relative to the stability of the enzymatic solution after 24 h of incubation at different pH at 30 ºC. Finally, SDS-PAGE analysis was removed in the revised manuscript.

  • The authors have mainly performed enzymatic activity probably on crude extract which is not clear How it was prepared. They have tested different pH, different temperatures and stability of xylanase but it is not clear How these experiments were performed and why the authors have not purified the xylanase to test these parameters which is a usual method for enzyme characterization?

RESPONSE: Thank you for your valuable observations. The enzymatic activity was performed in the fermentation supernatant after filtration and centrifugation. This information was added in the revised manuscript in section 2.6, lines 132-133. The objective of the authors by testing the non-purified enzyme was to demonstrate the potential properties and characteristics to indicate the potential of industrial applications. Since enzyme purification usually consists in time consuming and expensive methods and procedures, the authors believed it would not be necessary to achieve the objective of this specific study. Also, different applications require different levels of purification, and the possibility of applying an efficient enzyme with lower purification costs may contribute in lowering the costs of the industrial application of enzymes.

  • Line 192 to 195: The authors mentioned “The presence of non-quantified specific minor nutrients ………..” How the authors reached to this conclusion? Did they perform any experiment for the same?

RESPONSE: Since agro-industrial byproducts are complex mixtures of components obtained by its agricultural source it is expected to present several minor nutrients. A further in-deep characterization of these compounds was not performed since the analysis are time consuming and not cost-effective for the objective of the study. The authors believe a further characterization of these by-products would not contribute in a significative way for the manuscript since other composition of the agro-industrial by-products can also be found in the literature. Here the characterization was performed in order to evaluate the main components and its influence in the fungal cultivation.  

  • Line 202 to 207: Which results the authors are trying to show for mentioned text? Does central condition is the term used in literature or authors used it for the first time as it seems inappropriate.

RESPONSE: The sentence was rewritten and authors have specified in the revised manuscript that the central condition refers to the central point of the experimental design.

  • Line 226-229: The authors ended the statements abruptly and continued later as part of different results.

RESPONSE: Thank you for the observation, the sentence was rewritten in order to make it the reading more fluid.

  • Xylanase activity is higher at high temperature. Does it mean it can be stored at high temperature? Because it appears xylanase should have high stability in that case.

RESPONSE: Besides the high activity value presented at high temperature it is not directly correlated with the stability at the same temperature storage. The stability at different temperatures can be clearly observed in Figure 5B.

  • Line 296- 298: Not clear what authors want to convey and SDS is not informative.

RESPONSE: As mentioned before, SDS-PAGE analysis was removed from the manuscript.

  • I will recommend the authors to take help from native English speaker for data interpretation and presentation.

RESPONSE: Thank you for your recommendation. The authors believe that after the revision and the valuable comments, corrections and suggestions of the three referees the revised manuscript present an improved language and presentation, and is now suitable for publication.  

Round 2

Reviewer 3 Report

The authors have attempted to answer most of the questions raised by me in previous review to my statisfaction. 

However, I would have appreciated to include some of the advanced methods (as also pointed by other reviewers such as Zymograms or protein purification and its characterization) to support the claims made by the authors.  Removing the SDS image is diluting the information provided in the original submission. I will suggest to provide an improved quality of SDS image instead of completely removing it if authors does not want to include other methods.

There are still some grammatical errors that needs to be eliminated.

Author Response

Reviewer #3:

The authors have attempted to answer most of the questions raised by me in previous review to my statisfaction.

However, I would have appreciated to include some of the advanced methods (as also pointed by other reviewers such as Zymograms or protein purification and its characterization) to support the claims made by the authors.  Removing the SDS image is diluting the information provided in the original submission. I will suggest to provide an improved quality of SDS image instead of completely removing it if authors does not want to include other methods.

There are still some grammatical errors that needs to be eliminated.

RESPONSE: We thank the reviewer by the suggestions. As the reviewers 1 and 2 in the first revision suggested to remove the SDS image, we included the SDS image as a supplementary file in the revised manuscript. As suggested by the reviewer further purification will be performed in future experiments. Regarding the grammatical errors, the manuscript was revised and some grammatical errors were eliminated. Thank you again for your revision.